# STAT1 signaling shields T cells from NK cell-mediated cytotoxicity

Yu Hui Kang [ID] [1,2], Amlan Biswas[1,2,3], Michael Field [ID] [1,2] & Scott B. Snapper[1,2]

The JAK-STAT pathway critically regulates T-cell differentiation, and STAT1 is postulated to regulate several immune-mediated diseases by inducing proinflammatory subsets. Here we show that STAT1 enables CD4$^+$ T-cell-mediated intestinal inflammation by protecting them from natural killer (NK) cell-mediated elimination. $Stat1^{-/-}$ T cells fail to expand and establish colitis in lymphopenic mice. This defect is not fully recapitulated by the combinatorial loss of type I and II IFN signaling. Mechanistically, $Stat1^{-/-}$ T cells have reduced expression of $Nlrc5$ and multiple MHC class I molecules that serve to protect cells from NK cell-mediated killing. Consequently, the depletion of NK cells significantly rescues the survival and spontaneous proliferation of $Stat1^{-/-}$ T cells, and restores their ability to induce colitis in adoptive transfer mouse models. $Stat1^{-/-}$ mice however have normal CD4$^+$ T cell numbers as innate STAT1 signaling is required for their elimination. Overall, our findings reveal a critical perspective on JAK-STAT1 signaling that might apply to multiple inflammatory diseases.

[1] Division of Gastroenterology, Hepatology and Nutrition, Boston Children's Hospital, Boston, MA 02115, USA. [2] Harvard Medical School, Boston, MA 02115, USA. [3]Present address: Discovery Immunology, Abbvie, 200 Sidney Street, Cambridge, MA 02139, USA. Correspondence and requests for materials should be addressed to S.B.S. (email: Scott.Snapper@childrens.harvard.edu)

The JAK-STAT signaling pathway plays a critical role in transducing signals from various cytokines to achieve distinct transcriptional outcomes[1]. In T cells, this pathway has been well studied in terms of their regulation of T-cell differentiation[2]. Among the seven mammalian signal transducer and activator of transcription (STAT) family members, STAT1 is known to be important for the induction of Th1 cells downstream of IFNγ due to its induction of the transcription factor T-bet[3,4]. STAT1 has also been shown to suppress regulatory T-cell differentiation[5]. These proinflammatory properties of STAT1 are important for controlling infections, where patients with loss-of-function mutations in *Stat1* develop susceptibility to viral/mycobacterial infections[6]. They are also important for promoting inflammatory diseases like graft-vs-host-disease (GvHD)[5]. However, STAT1 also suppresses Th17 differentiation[7], and *Stat1*$^{-/-}$ mice develop aggravated Th17-mediated autoimmune diseases including experimental autoimmune encephalomyelitis (EAE)[8,9].

Inflammatory bowel diseases (IBD) likely arise from an aberrant immune response toward intestinal microbes in a genetically susceptible host[10]. Crohn's disease in particular is characterized by a skewing of the CD4$^+$ T cell profile toward the proinflammatory Th1 and Th17 subsets, which are believed to be critical for disease pathogenesis[11]. Patients with Crohn's disease display higher STAT1 expression, albeit only modestly in CD4$^+$ T cells[12]. However, the mechanism by which STAT1 modulates CD4$^+$ T cells in IBD is currently unclear and presumed to be through altering differentiation states[13].

IL-10 is a critical anti-inflammatory cytokine for maintaining intestinal immune homeostasis, as evidenced in mice and humans deficient in IL-10 or IL-10 receptor (IL-10R) that develop spontaneous colitis[14–16]. We and others recently described the importance of IL-10R signaling in macrophages in the prevention of colitis, with *Il10rb*$^{-/-}$*Rag1*$^{-/-}$ mice but not *Rag1*$^{-/-}$ mice developing colitis upon reconstitution with WT CD4$^+$ T cells[17,18]. Subsequent studies in our model and others pointed to a role for pathogenic Th17 cells in driving the disease[19–24]. As STAT1 is a critical regulator of Th1/Th17 differentiation, we further investigated its role in the ability of CD4$^+$ T cells to induce colitis.

Here we describe a role for STAT1 in enabling T cells to induce colitis by protecting them from NK cell-mediated cytotoxicity. *Stat1*$^{-/-}$ T cells fail to expand and induce colitis in vivo unless NK cells are depleted. This is because STAT1 is required to induce sufficient levels of *Nlrc5* and the inhibitory NK ligand MHC class I to enable evasion of rejection by host NK cells. Surprisingly, this requirement for STAT1 is largely independent of both Type I and II IFN signaling, the classical activators of STAT1. Moreover, this mechanism is specific to *Stat1*$^{-/-}$ T cells undergoing spontaneous proliferation and requires STAT1 expression in the innate compartment. Altogether, our study reveals a critical role of STAT1 that is distinct from T-cell differentiation and adds a new perspective to studies on T-cell-mediated inflammatory disease.

## Results

**T cells require STAT1 to expand and induce colitis in vivo**. To investigate the role of STAT1 signaling in T-cell driven colitis, we adoptively transferred unfractionated WT or *Stat1*$^{-/-}$ CD4$^+$ T cells into *Il10rb*$^{-/-}$*Rag1*$^{-/-}$ mice (Fig. 1a). WT T cells induced severe colitis in *Il10rb*$^{-/-}$*Rag1*$^{-/-}$ recipient mice as expected[17]. In contrast, mice transferred with *Stat1*$^{-/-}$ T cells displayed no signs of intestinal inflammation as evidenced by the lack of weight loss, colonic thickening and histological inflammation (Fig. 1a, b). Flow cytometric analysis of the colonic lamina propria revealed a marked reduction of *Stat1*$^{-/-}$ T cells compared to

WT T cells (Fig. 1c). This was not due to aberrant homing of *Stat1*$^{-/-}$ T cells to the intestine, as a similar reduction of T cells was observed in the spleen (Fig. 1d).

We next asked if the reduction of *Stat1*$^{-/-}$ T cells was dependent on colonic inflammation by transferring unfractionated WT or *Stat1*$^{-/-}$ CD4$^+$ T cells into *Rag1*$^{-/-}$ mice, a strain that does not develop colitis when reconstituted with unfractionated WT T cells[17]. Similar to *Il10rb*$^{-/-}$*Rag1*$^{-/-}$ mice, *Stat1*$^{-/-}$ T cells were markedly reduced in the colons and spleens of *Rag1*$^{-/-}$ mice, indicating that STAT1 is required for robust in vivo T-cell expansion independent of colonic inflammation and innate IL-10R expression (Fig. 2a, b).

**Partial dependency of STAT1 on Type I + II IFN signaling**. IFNs are the classical inducers of STAT1 signaling with both Type I and Type II IFN individually reported to regulate T cell function[3,25–27]. We therefore sought to determine if the impaired expansion of *Stat1*$^{-/-}$ T cells was due to the lack of both type I and type II IFN signaling by transferring *Ifnar1*$^{-/-}$*Ifngr1*$^{-/-}$ CD4$^+$ T cells into *Il10rb*$^{-/-}$*Rag1*$^{-/-}$ mice (Fig. 3a). Surprisingly, the abrogation of both Type I and Type II IFN receptors failed to recapitulate STAT1 deficiency, as *Ifnar1*$^{-/-}$*Ifngr1*$^{-/-}$ CD4$^+$ T cells expanded to similar levels as WT T cells in the spleen and colon 3 weeks post transfer (Fig. 3a).

*Ifnar1*$^{-/-}$*Ifngr1*$^{-/-}$ T cells were also able to induce colitis unlike *Stat1*$^{-/-}$ T cells (Fig. 3b, c). However, the severity of colitis induced by *Ifnar1*$^{-/-}$*Ifngr1*$^{-/-}$ T cells was reduced compared to WT T cells (Fig. 3b, c), which correlated with a reduced rate of expansion of *Ifnar1*$^{-/-}$*Ifngr1*$^{-/-}$ T cells in the blood (Fig. 3d). As expected, *Stat1*$^{-/-}$ T cells did not expand in the blood (Fig. 3d). These data suggest that while Type I+II IFN partially contribute to the STAT1-dependent signaling, the impaired expansion of *Stat1*$^{-/-}$ T cells is predominantly an IFN-independent process at later time points.

**Cell-intrinsic role for STAT1 in in vivo T-cell expansion**. To understand the mechanisms linking STAT1 to T-cell expansion in vivo, we first asked if the presence of WT T cells could rescue the defective expansion of *Stat1*$^{-/-}$ T cells by transferring equal ratios of congenically marked WT (CD45.1$^+$) and *Stat1*$^{-/-}$ (CD45.2$^+$) T cells into *Rag1*$^{-/-}$ mice (Fig. 4a). Notably, all of the T cells identified three weeks post transfer were WT, indicating that the defective expansion of *Stat1*$^{-/-}$ T cells is cell-intrinsic (Fig. 4b).

We next asked if the defective expansion of *Stat1*$^{-/-}$ T cells could be recapitulated in vitro. In contrast with the in vivo defect, *Stat1*$^{-/-}$ T cells displayed a hyperproliferative phenotype compared to WT T cells upon in vitro stimulation (Supplementary Fig. 1), consistent with earlier reports[5,28]. This indicates that the expansion defect of *Stat1*$^{-/-}$ T cells is not cell autonomous and requires an in vivo environment.

**Reduced expression of the MHC-I pathway in *Stat1*$^{-/-}$ T cells**. To investigate whether a dysregulated transcriptional profile might account for the observed defect, we performed gene expression analysis on *Stat1*$^{-/-}$ T cells pre and post transfer into *Rag1*$^{-/-}$ mice by RNA-seq. As *Stat1*$^{-/-}$ T cells failed to expand to appreciable amounts after 3 weeks, we transferred a larger number of cells and analyzed gene expression at 1 week post transfer, a time point where *Stat1*$^{-/-}$ T cells were beginning to decline (Fig. 5a and Supplementary Fig. 2a). Gene ontology (GO) analysis of genes differentially regulated between WT and *Stat1*$^{-/-}$ T cells revealed, as expected, categories related to Type I and II IFN signaling. Interestingly, categories related to the MHC class I (MHC-I) antigen presentation pathway were significantly

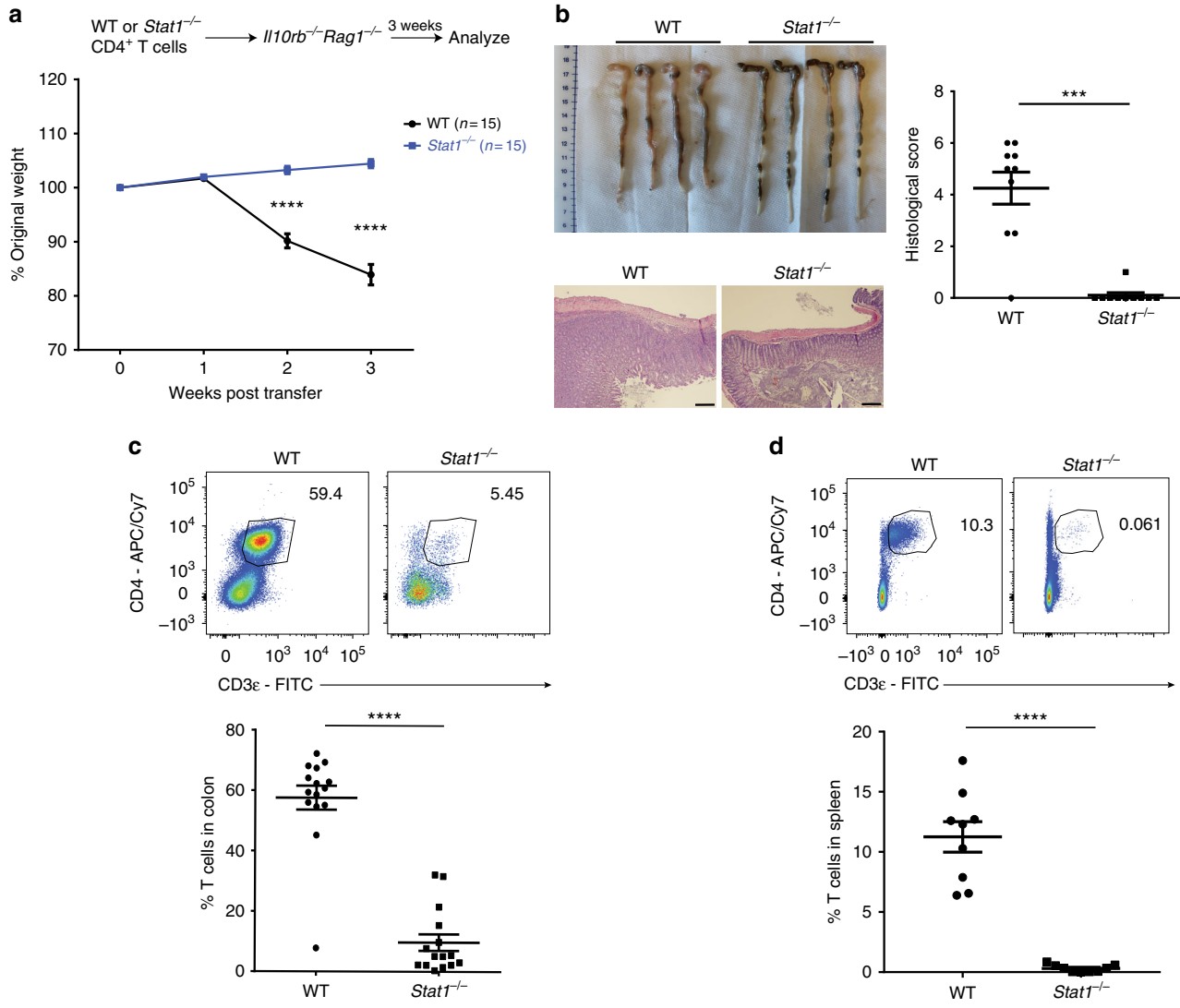

**Fig. 1** *Stat1*$^{-/-}$ T cells fail to induce colitis due to defective expansion. *Il10rb*$^{-/-}$*Rag1*$^{-/-}$ mice were injected i.p. with $1 \times 10^6$ unfractionated WT or *Stat1*$^{-/-}$ CD4$^+$ T cells. **a** Mean % original body weights ± SEM following T-cell transfer. Source data are provided as a Source Data file. **b** Representative images of colons, as well as representative H&E images of distal colon sections with mean histological scores ± SEM at 3 weeks post transfer. Scale bar represents 200 μm. **c, d** Representative flow cytometry plots of CD4$^+$ T cells (gated on live CD45$^+$ cells, Supplementary Fig. 4a) in the **c** colon and **d** spleen followed by their mean frequencies ± SEM at 3 weeks post transfer. All data are pooled from two to three independent experiments, with each point representing an individual mouse. ****$p < 0.0001$, ***$p < 0.001$ by **a** two-way ANOVA with Bonferroni's correction or **b**–**d** two-tailed Mann–Whitney test

enriched in both pre and post-transfer settings (Fig. 5a and Supplementary Fig. 2b). Consistent with the GO analysis, *Stat1*$^{-/-}$ T cells had reduced expression of *Nlrc5*, MHC-I (*H2-K1, H2-D1, B2m, H2-T23*) and various genes involved in MHC-I antigen presentation (*Tap1, Tap2, Psmb8, Psmb9*) (Fig. 5b and Supplementary Fig. 2c).

NLRC5 is a critical transactivator of multiple MHC-I genes, and STAT1 is required to induce its expression by binding to the *Nlrc5* promoter in response to IFNγ[29–32]. Consistent with our RNA-seq data and with earlier reports[31,33], *Stat1*$^{-/-}$ T cells displayed reduced surface levels of the classical MHC-I molecules H-2K$^b$/H-2D$^b$. Interestingly, surface levels of the non-classical molecule Qa-1 was only mildly affected by STAT1 deficiency whereas levels of Qa-2 were severely reduced (Fig. 5c).

**NK depletion rescues *Stat1*$^{-/-}$ T-cell expansion and colitis.** The MHC-I molecule is the classic inhibitory ligand for NK cells, and

cellular expression of MHC-I protects cells from NK mediated killing[34]. Tumors or virally infected cells can reduce MHC-I expression to evade CD8$^+$ T-cell recognition, but this renders them susceptible to NK mediated killing—a phenomenon described as missing self[34,35]. The reduced expression of MHC-I on *Stat1*$^{-/-}$ T cells led us to hypothesize that their defective expansion was due to elimination by NK cells, which are present and more active in *Rag1*$^{-/-}$ mice[36]. This hypothesis was supported by the GO analysis, which revealed the category: Protection from natural killer mediated cytotoxicity (Fig. 5a and Supplementary Fig. 2b).

To test the hypothesis that *Stat1*$^{-/-}$ T cells were eliminated in vivo by NK cells, we depleted NK cells in *Rag1*$^{-/-}$ mice at the time of T-cell transfer by employing an anti-NK1.1 antibody (Fig. 6a and Supplementary Fig. 3a). In support of the hypothesis, the depletion of NK cells significantly rescued the survival of *Stat1*$^{-/-}$ T cells (Fig. 6a). We next asked if *Stat1*$^{-/-}$ T cells were able to induce colitis in the absence of NK cells by transferring

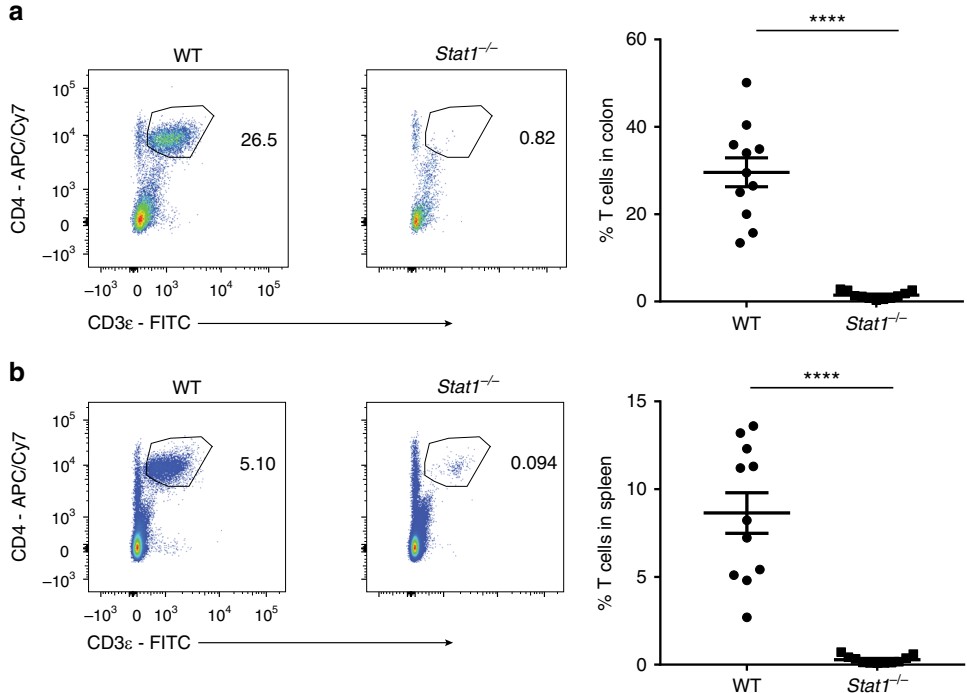

**Fig. 2** *Stat1*$^{-/-}$ T cells fail to expand in *Rag1*$^{-/-}$ mice. *Rag1*$^{-/-}$ mice were injected i.p. with $1 \times 10^6$ WT or *Stat1*$^{-/-}$ unfractionated CD4$^+$ T cells and analyzed 3 weeks post transfer. **a, b** Representative flow cytometry plots of CD4$^+$ T cells (gated on live CD45$^+$ cells) in the colon (**a**) and spleen (**b**) followed by their mean frequencies ± SEM. Data are pooled from three independent experiments, with each point representing an individual mouse. ****$p <$ 0.0001 by two-tailed Mann–Whitney test

them into NK cell-depleted *Il10rb*$^{-/-}$*Rag1*$^{-/-}$ mice. Strikingly, *Stat1*$^{-/-}$ T cells were able to induce disease in NK cell-depleted *Il10rb*$^{-/-}$*Rag1*$^{-/-}$ mice unlike their control-treated NK cell-replete counterparts (Fig. 6b, c). The induction of disease correlated with a restored expansion of *Stat1*$^{-/-}$ T cells in the spleen and the colon (Fig. 6d). In agreement with previous reports[7], *Stat1*$^{-/-}$ T cells displayed an enhanced Th17 differentiation profile in vivo (Supplementary Fig. 3b). This differentiation profile was however seen in both control and NK cell-depleted *Il10rb*$^{-/-}$*Rag1*$^{-/-}$ mice, suggesting that the primary role of STAT1 in T-cell-driven colitis is to protect the T cells from NK mediated elimination, rather than to repress their intrinsic Th17 differentiation potential.

**NK cells target spontaneously proliferating *Stat1*$^{-/-}$ T cells.** T cells undergo two distinct modes of proliferation upon transfer into chronically lymphopenic hosts (e.g., *Rag1*$^{-/-}$ mice)—slow, true homeostatic proliferation (HP) that is driven primarily by IL-7, as well as rapid, spontaneous proliferation (SP) that is driven by the microbiota and IL-6[37–41]. We asked whether the NK cell-mediated elimination of *Stat1*$^{-/-}$ T cells requires T-cell proliferation by transferring CellTrace Violet (CTV) labeled WT or *Stat1*$^{-/-}$ CD4$^+$ T cells into *Rag1*$^{-/-}$ mice. *Stat1*$^{-/-}$ T cells displayed a reduction in the SP population compared to WT T cells, with no difference in the HP population (Fig. 7a). Importantly, the depletion of NK cells significantly rescued the *Stat1*$^{-/-}$ SP population, suggesting that NK cells specifically restrict *Stat1*$^{-/-}$ T cells undergoing SP (Fig. 7b). Interestingly, this rescue was not complete as we also observed an increase in the SP of WT T cells upon NK cell depletion, which is consistent with the incomplete rescue of *Stat1*$^{-/-}$ T-cell expansion as well as the degree of colitis induced in *Il10rb*$^{-/-}$*Rag1*$^{-/-}$ mice (Fig. 6).

We also assessed for cell death in these populations by staining for activated caspases using FAM-FLICA, a fluorescently conjugated pan-caspase inhibitor. Compared to WT T cells,

*Stat1*$^{-/-}$ T cells displayed increased cell death specifically in the SP population, with no differences in the HP or non-proliferating populations (Fig. 7c). Importantly, the increased cell death in *Stat1*$^{-/-}$ SP T cells was reversed by NK cell depletion (Fig. 7d). Taken together, these data strongly suggest that NK cells eliminate *Stat1*$^{-/-}$ T cells when they undergo SP.

**Innate STAT1 expression is needed to reject *Stat1*$^{-/-}$ T cells.** Despite the potent elimination of *Stat1*$^{-/-}$ T cells upon adoptive transfer into lymphopenic hosts, *Stat1*$^{-/-}$ mice had normal levels of CD4$^+$ T cells, suggesting additional mechanism(s) in place to prevent their elimination by NK cells (Fig. 8a). As STAT1 is required for NK cells to achieve optimal cytotoxicity[42,43], we hypothesized that these T cells were not eliminated in *Stat1*$^{-/-}$ mice due to a defect in killing by *Stat1*$^{-/-}$ NK cells. To test this hypothesis, we deleted *Stat1* in the innate compartment by generating *Stat1*$^{-/-}$*Rag1*$^{-/-}$ mice and transferred congenically marked WT (CD45.1$^+$) and *Stat1*$^{-/-}$ (CD45.2$^+$) T cells into them. Whereas *Stat1*$^{-/-}$ T cells were efficiently depleted in the *Stat1*$^{+/-}$ *Rag1*$^{-/-}$ littermate controls, deletion of *Stat1* in the innate compartment restored the expansion of *Stat1*$^{-/-}$ T cells (Fig. 8b). This indicates that the elimination of *Stat1*$^{-/-}$ T cells is dependent on innate STAT1 signaling.

## Discussion

In this study we have identified a critical role for STAT1 in T-cell survival, where STAT1 signaling, through the upregulation of *Nlrc5* and MHC-I, protects T cells from NK cell-mediated elimination in vivo. We also show that this is important in the setting of T-cell-mediated immunopathology, as *Stat1*$^{-/-}$ T cells can induce colitis if allowed to survive and expand in a NK-deficient environment.

In T cells, most studies on the JAK-STAT pathway have focused on its effects on T-cell differentiation, with STAT1

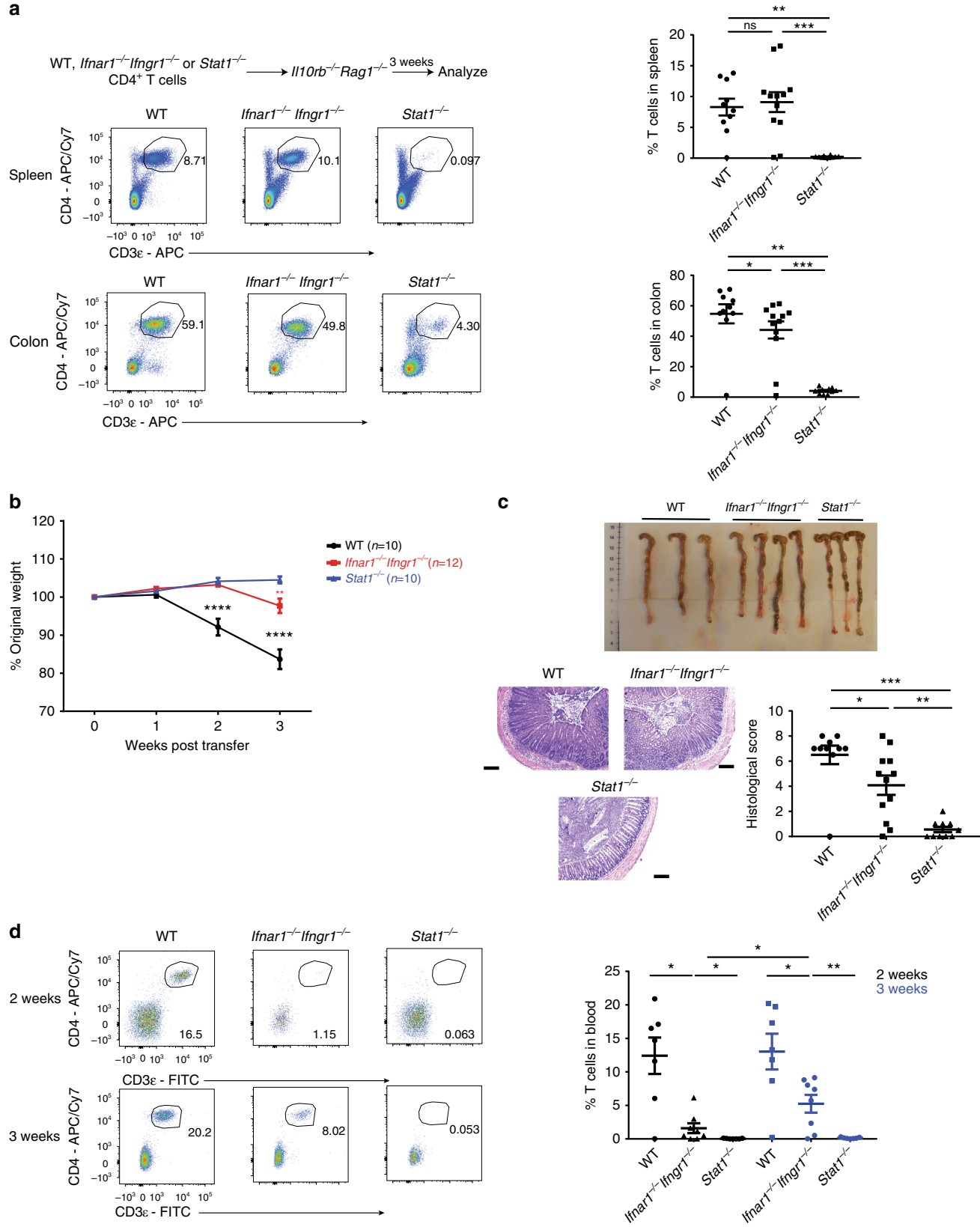

promoting Th1 differentiation (through the induction of T-bet) and inhibiting Th17 differentiation[3,4,7]. In IBD, previous studies on STAT1 signaling in T cells focused on the STAT1-dependent transcription factor T-bet[13,23]. While it was noted that $Stat1^{-/-}$ T cells were unable to cause colitis, the profile of $Stat1^{-/-}$ T cells in vivo was not analyzed and STAT1 was assumed to act in a similar fashion as T-bet[13]. We observe that in our model of colitis[17], STAT1 modulates the disease outcome primarily by promoting T-cell survival rather than altering differentiation, as $Stat1^{-/-}$ T cells displayed similar

**Fig. 3** Type I + II IFN signaling do not explain the defective expansion of $Stat1^{-/-}$ T cells. $Il10rb^{-/-}Rag1^{-/-}$ mice were injected i.p. with $1 \times 10^6$ WT, $Ifnar1^{-/-}Ifngr1^{-/-}$ or $Stat1^{-/-}$ CD4$^+$ T cells. **a** Representative flow cytometry plots of CD4$^+$ T cells (gated on live CD45$^+$ cells, Supplementary Fig. 4b) in the spleen and colon followed by their mean frequencies ± SEM at 3 weeks post transfer. **b** Mean % initial body weights ± SEM following T-cell transfer. Source data are provided as a Source Data file. **c** Representative images of colons, as well as representative H&E images of distal colon sections with mean histological scores ± SEM at 3 weeks post transfer. Scale bar represents 200 μm. **d** Representative flow cytometry plots of CD4$^+$ T cells (gated on CD45$^+$ cells) in the blood followed by their mean frequencies ± SEM at 2 and 3 weeks post transfer. All data are pooled from two to three independent experiments, with each point representing an individual mouse. $*p < 0.05$, $**p < 0.01$, $***p < 0.001$, $****p < 0.0001$ by **b** two-way ANOVA with Bonferroni's correction (WT compared to $Ifnar1^{-/-}Ifngr1^{-/-}$ or $Stat1^{-/-}$, $Ifnar1^{-/-}Ifngr1^{-/-}$ compared to $Stat1^{-/-}$) or by **a, c, d** two-tailed Mann–Whitney test

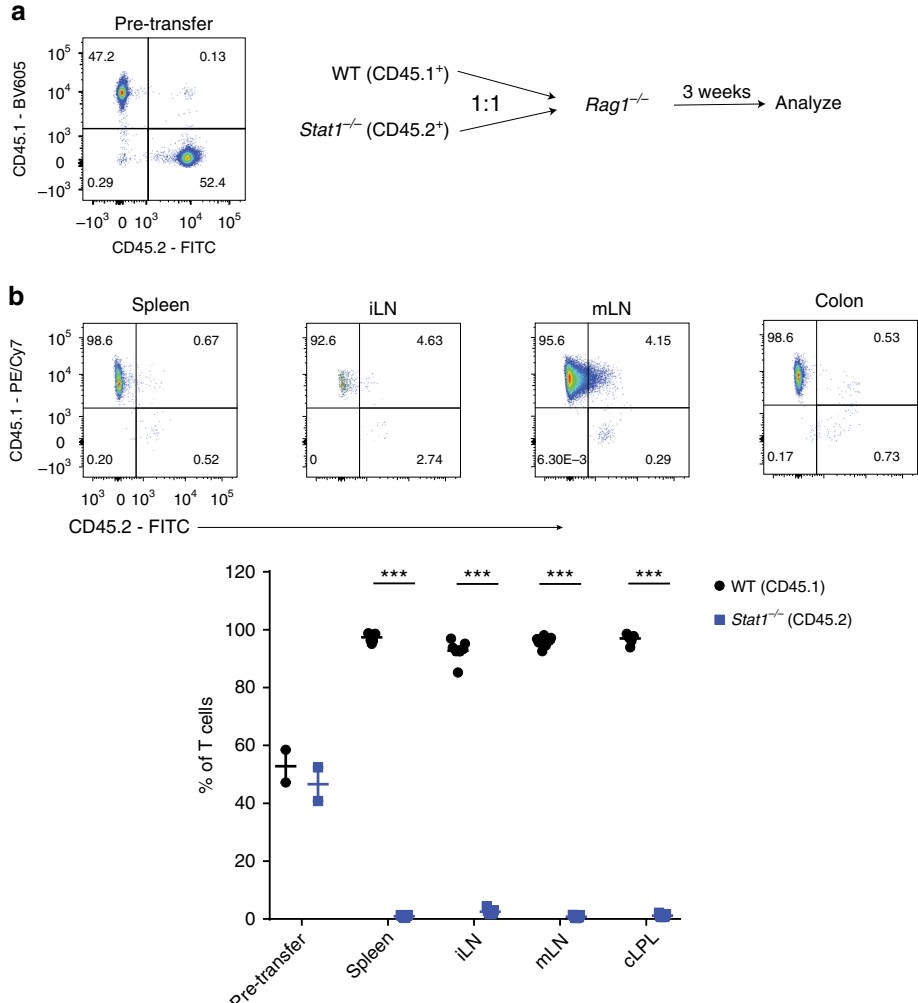

**Fig. 4** The defective expansion of $Stat1^{-/-}$ T cells is cell-intrinsic. WT (CD45.1$^+$) or $Stat1^{-/-}$ (CD45.2$^+$) unfractionated CD4$^+$ T cells were injected i.p. at a 1:1 ratio (0.8–1 × 10$^6$/type) into $Rag1^{-/-}$ mice. **a** Schematic of experiment and representative plot of cells injected. **b** Representative images of CD45.1$^+$ vs CD45.2$^+$ cells (gated on live CD45$^+$ CD3ε$^+$ CD4$^+$ T cells) from various organs followed by their mean frequencies ± SEM. Data is pooled from two independent experiments, with each point representing an individual mouse. $***p < 0.001$ by two-tailed Mann–Whitney test. Accompanied by Supplementary Fig. 1

differentiation profiles in both control and NK-depleted $Il10rb^{-/-}Rag1^{-/-}$ hosts (Supplementary Fig. 3b).

While we have focused mainly on IBD, we believe that the STAT1-mediated protection from NK cell killing is relevant for several T-cell-mediated inflammatory disorders. STAT1 induces MHC-I expression by inducing the transcription of $Nlrc5$[29–32], and deletion of $Nlrc5$ in T cells to reduce MHC-I expression leads to their rejection by host NK cells during viral infection[44]. In the case of STAT1 deficiency, while several groups have reported a reduced expansion of $Stat1^{-/-}$ T cells in animal models of autoimmune disease and GvHD, the mechanistic

basis for this reduction was unclear[5,9]. We propose that, similar to IBD, NK cells might also play a prominent role in restraining the ability of $Stat1^{-/-}$ T cells to induce these inflammatory disorders. This is analogous to studies in tumor biology, where tumor cells utilize STAT1 expression to avoid rejection in vivo[43]. Our finding that only $Stat1^{-/-}$ T cells undergoing SP are targeted by NK cells suggests that NK cells restrict T cells only when they are activated (Fig. 7). This is in agreement with earlier studies where $Ifnar1^{-/-}$ antiviral T cells are only eliminated by NK cells when the mice are virally infected[26,45] and suggests that a similar mechanism might be used to

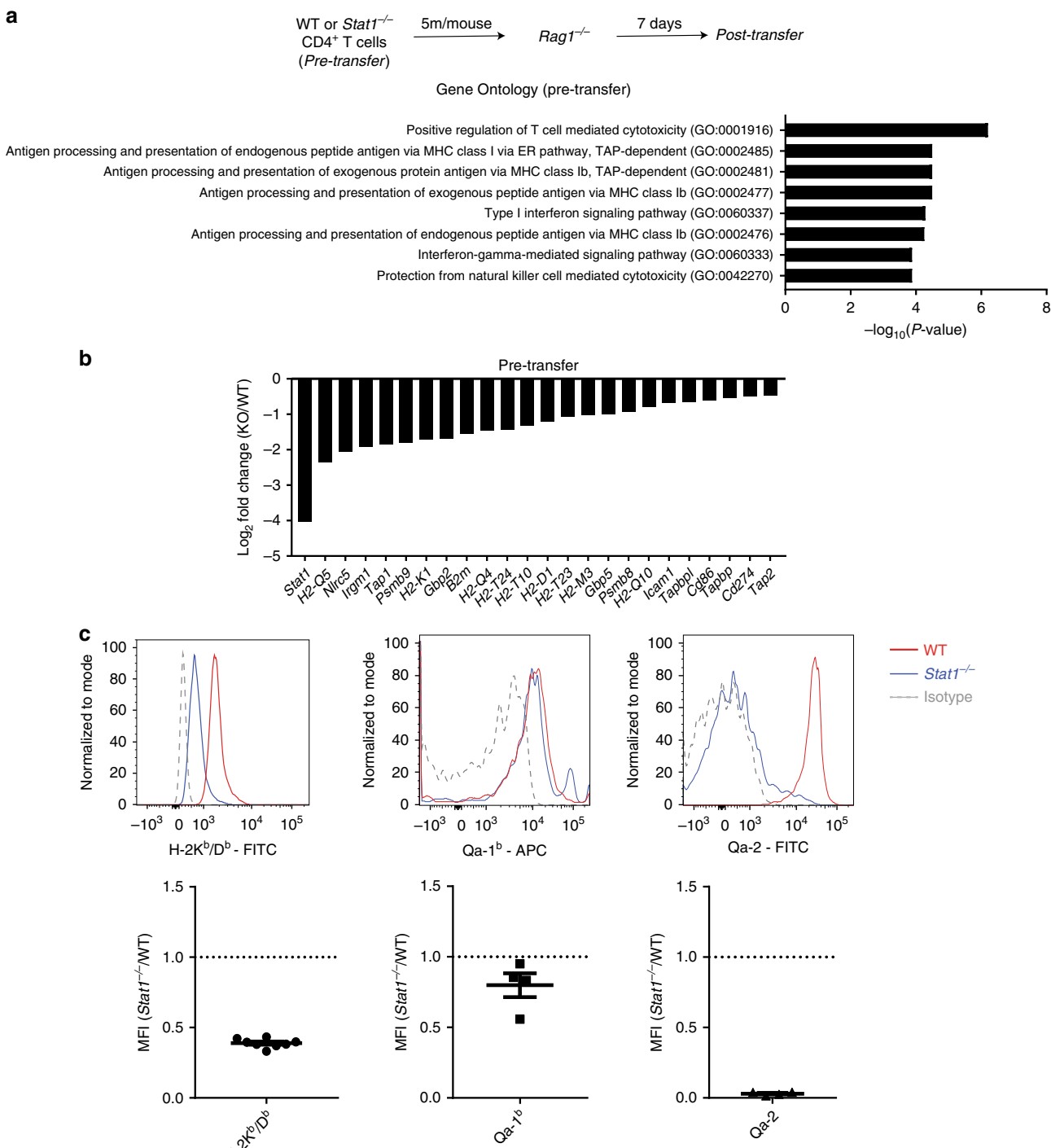

**Fig. 5** Downregulation of the MHC class I antigen presentation machinery in *Stat1*⁻/⁻ T cells. RNA-seq was performed on WT or *Stat1*⁻/⁻ T cells pre-transfer. **a** Schematic of experimental setup and selected Gene ontology terms (PANTHER) showing differential expression of the MHC class I pathway in *Stat1*⁻/⁻ T cells. **b** Downregulation of various genes involved in MHC class I antigen presentation in *Stat1*⁻/⁻ T cells compared to WT T cells by RNA-seq. All genes displayed are significantly different between WT and *Stat1*⁻/⁻ T cells ($n = 3$ biological replicates, $p < 0.05$ with correction for multiple testing by Benjamini-Hochberg procedure). Accompanied by Supplementary Fig. 2 where similar analyses were performed in T cells post transfer. **c** Representative flow cytometry plots showing surface expression of classical and non-classical MHC class I molecules on CD4⁺ T cells from the spleens of WT or *Stat1*⁻/⁻ mice, followed by their cumulative enumeration expressed as a ratio of Median Fluorescence Intensity (*Stat1*⁻/⁻ / WT) ± SEM. Data is pooled from three or more independent experiments, with each point representing an individual mouse. Similar numbers of WT and *Stat1*⁻/⁻ mice were used for the comparison

regulate commensal-driven T-cell responses. Recently, it was reported that overexpression of STAT1 in T cells inhibits their expansion in lymphopenic mice, which led to the suggestion of targeting STAT1 to enhance T-cell numbers in clinical settings of lymphopenia like bone marrow transplantation and HIV infection[46]. Our findings suggest that this approach will have to be balanced with ensuring that there is sufficient MHC-I expression to protect T cells from being targeted by NK cells.

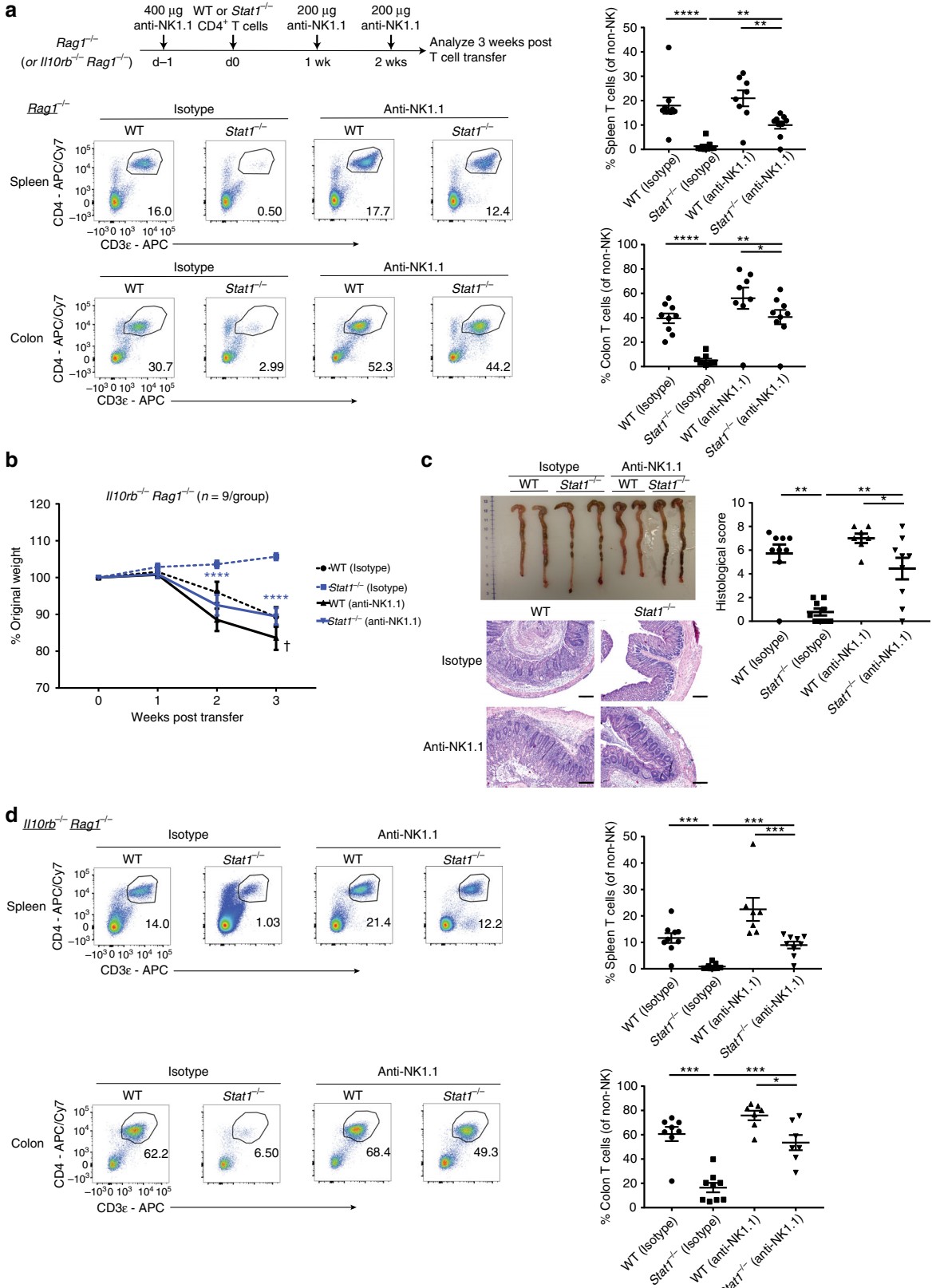

The upstream signal(s) that activates the STAT1-NLRC5-MHC class I axis in T cells in vivo has not been fully elucidated. In vitro, *Nlrc5* expression in T cells is primarily triggered by autocrine IFNγ signaling[31]. In vivo, type I IFN has been reported to protect antiviral T cells and NK cells from NK mediated elimination during LCMV infection[26,45,47]. Our data stands in contrast with these studies, showing that deletion of both Type I and Type II IFN receptors fails to fully recapitulate the defective survival of *Stat1*[−/−] T cells in the setting of IBD (Fig. 3). This is consistent with an earlier report showing that type I IFN signaling is not

**Fig. 6** Depletion of NK cells restores $Stat1^{-/-}$ T-cell expansion and colitis. **a** $1 \times 10^6$ WT or $Stat1^{-/-}$ CD4$^+$ T cells were injected i.p. into $Rag1^{-/-}$ mice that were treated with NK depleting antibody (or isotype control). Schematic of experimental setup, as well as representative flow cytometry plots of CD4$^+$ T cells (gated on live CD45$^+$ non-NK cells, Supplementary Fig. 4c) in the spleen and colon of $Rag1^{-/-}$ mice with their mean frequencies ± SEM at 3 weeks post transfer. **b–d** Similar to **a**, but in $Il10rb^{-/-}Rag1^{-/-}$ mice instead of $Rag1^{-/-}$ mice. **b** Mean % initial body weights ± SEM following T-cell transfer. Source data are provided as a Source Data file. **c** Representative images of colons, as well as representative H&E images of distal colon sections with mean histological scores ± SEM at 3 weeks post transfer. **d** Representative flow cytometry plots of CD4$^+$ T cells (gated on live CD45$^+$ non-NK cells, Supplementary Fig. 4c) in the spleen and colon followed by their mean frequencies ± SEM at 3 weeks post transfer. Scale bar represents 200 μm. Data pooled from three independent experiments. *$p < 0.05$, **$p < 0.01$, ***$p < 0.001$, ****$p < 0.0001$ by **b** two-way ANOVA with Bonferroni's correction ($Stat1^{-/-}$ anti-NK1.1 compared to $Stat1^{-/-}$ Isotype) or by **a**, **c**, **d** two-tailed Mann–Whitney test. † Two mice (WT anti-NK1.1) were sacrificed before the 3-week time point due to excessive weight loss thus their weights only apply till week 2. Accompanied by Supplementary Fig. 3

required for naïve T cells to induce colitis in $Rag^{-/-}$ hosts[48], but we further extend this observation to include Type II IFN signaling. What are the IFN-independent signals that might account for the discrepancy between $Ifnar1^{-/-}Ifngr1^{-/-}$ and $Stat1^{-/-}$ T cells? IL-7 has been proposed as a possible candidate, being able to induce STAT1 activation in T cells in vitro and in vivo[33,46] as well as MHC-I in vitro[33]. Therefore, in addition to the conventional STAT5-driven proliferative and pro-survival response[49], IL-7 might activate STAT1 signaling to induce MHC-I for protection from NK cells. However, our finding that NK cells specifically eliminate $Stat1^{-/-}$ T cells undergoing SP and not the IL-7 driven HP argues against this hypothesis (Fig. 7). IL-6, which has been reported to be important in driving SP, also activates STAT1[40,41,50]. However, a recent report showed that IL-6R deficient T cells only display defective expansion in $Rag1^{-/-}$ mice when there is colonic inflammation[51], which is in contrast to our observations with $Stat1^{-/-}$ T cells (Fig. 2). It is possible that the STAT1-dependent signal is provided by multiple cytokines, including IL-6 and type I+II IFN. Alternatively, the maintenance of MHC-I levels might be driven by tonic STAT1 signaling that is independent of any upstream cytokine engagement.

The ability of NK cells to restrict T-cell expansion has been noted in mouse models of infection[52] and IBD[53,54], and our data and others[26,45] suggest that STAT1 signaling plays a critical role in this regulation. However, the extent of STAT1 dependency might be different in different disease contexts. While $Stat1^{-/-}$ T cells are very efficiently eliminated in our study and others[9], the degree of reduction in expansion of these T cells are not as pronounced in GvHD[5]. Moreover, in a recent report employing a mouse model of EAE, $Stat1^{-/-}$ T cells induced worse disease than WT T cells, although it is unclear whether they had an expansion defect[55]. It is thus possible that in certain disease contexts the inflammatory environment can provide STAT1-independent signals to the T cells to protect them from NK cells. Alternatively, the NK cells might be altered in these settings toward reduced cytotoxicity.

We observe that $Stat1^{-/-}$ T cells fail to expand in both $Rag1^{-/-}$ and $Il10rb^{-/-}Rag1^{-/-}$ mice (Figs. 1, 2), suggesting that IL-10Rβ signaling is not required for NK cells to eliminate $Stat1^{-/-}$ T cells. However, earlier work has shown roles for IL-10 and IFNλ/IL-28—both of which signal via the IL-10Rβ chain—in stimulating NK cells[56–58]. It is possible that $Il10rb^{-/-}$ NK cells might still have sufficient cytotoxic ability to eliminate $Stat1^{-/-}$ T cells, as IL-28R deficient NK cells are only partially defective[58]. Alternatively, there might be other mechanisms in $Il10rb^{-/-}Rag1^{-/-}$ mice that compensate for this defect, such as IL-10R deficiency in macrophages which promotes a proinflammatory environment[17] and/or RAG1 deficiency which has been shown to lead to NK cell hyperresponsiveness[36].

In our study, we also show that innate STAT1 signaling is required to eliminate $Stat1^{-/-}$ T cells, which would explain why $Stat1^{-/-}$ mice have normal levels of T cells (Fig. 8). We believe that this is likely due to the impaired cytotoxic capability of

$Stat1^{-/-}$ NK cells as previously reported[42,43]. However, we do not exclude the possibility that this can also be due to altered NK education, where NK cells are educated to recognize the low level of MHC-I on $Stat1^{-/-}$ T cells as normal.

In summary, we describe a critical role for STAT1 in promoting T-cell survival by maintaining sufficient MHC class I expression to evade NK cell-mediated killing. This mechanism is largely IFN-independent and is critical in enabling T cells to induce intestinal inflammation. Our findings shed a new light on JAK-STAT signaling in T cells, adding critical functions for this pathway beyond T-cell differentiation that have potential therapeutic implications for IBD and other T-cell-mediated inflammatory disorders.

## Methods

**Mouse strains**. C57BL/6J (Strain 000664), B6.SJL-$Ptprc^a$ $Pepc^b$/BoyJ (CD45.1, Strain 002014), B6.129S7-$Rag1^{tm1Mom}$/J ($Rag1^{-/-}$, Strain 002216), B6.129S(Cg)-Stat1$^{tm1Dlv}$ ($Stat1^{-/-}$, Strain 012606), B6.Cg-$Ifngr1^{tm1Agt}Ifnar1^{tm1.2Ees}$/J ($Ifnar1^{-/-}Ifngr1^{-/-}$, Strain 029098) mice were purchased from Jackson Labs. $Il10rb^{-/-}Rag1^{-/-}$ mice were generated by crossing $Il10rb^{-/-}$ mice (a gift from Thaddeus Stappenbeck, Washington University) with $Rag1^{-/-}$ mice. $Stat1^{-/-}Rag1^{-/-}$ mice were generated by crossing $Stat1^{-/-}$ mice with $Rag1^{-/-}$ mice. All mice were on the B6 background and maintained in a specific pathogen-free animal facility in Boston Children's Hospital. All experiments were conducted after approval from the Animal Resources at Children's Hospital and according to regulations by the Institutional Animal Care and Use Committee (IACUC).

**Adoptive T-cell transfer and colitis induction**. In T-cell transfer experiments, unfractionated CD4$^+$ T cells were isolated from the spleens and lymph nodes of donor mice (WT, CD45.1, $Stat1^{-/-}$, $Ifnar1^{-/-}Ifngr1^{-/-}$) by negative selection (Miltenyi Biotec CD4$^+$ T-cell isolation Kit, Cat No. 130-104-454). $1 \times 10^6$ T cells (92.7–98.6% pure) were then adoptively transferred into recipient mice ($Rag1^{-/-}$, $Il10rb^{-/-}Rag1^{-/-}$) by i.p. injection in PBS unless otherwise stated. In some experiments, CD45.1$^+$ T cells and $Stat1^{-/-}$ T cells were mixed at a 1:1 ratio before being transferred into recipient mice ($Rag1^{-/-}$, $Stat1^{-/-}Rag1^{-/-}$). In some experiments, T cells were labeled with 5 μM CellTrace Violet (Thermo Fisher) in PBS + 0.1% FBS for 10 min at 37 °C prior to injection. All recipient mice were at least 6 weeks old and matched for sex, age, and housing between groups. $Il10rb^{-/-}Rag1^{-/-}$ mice were monitored weekly for body weight changes post T cell transfer. For NK depletion assays, each mouse was first injected with 400 μg anti-NK1.1 (or isotype control) 1 day prior to T-cell transfer. For experiments lasting beyond 1 week, depletion of NK cells was maintained by injections of 200 μg anti-NK1.1 (or isotype control) at 1 and 2 weeks post transfer. All antibody injections were administered i.p in InVivoPure pH 7.0 Dilution Buffer (BioXCell).

**Histological scoring**. To evaluate signs of histological inflammation, sections of distal colons were stained in haematoxylin and eosin and scored in a blinded fashion. Scoring was based on histological evidence of crypt hyperplasia (0–3), inflammatory cell infiltration (0–3) and presence of crypt abscesses (0–2), summed up to give the overall score (0–8). Representative images were acquired using an Olympus BX41 upright microscope with DP70 color CCD (Fig. 1) or a Keyence automated epifluorescent microscope (Figs. 3, 6).

**Isolation of colonic lamina propria cells**. Cells were isolated from the lamina propria as described[17]. Briefly, the large intestine (colon + cecum) was removed, cut open longitudinally and then into small sections before being incubated in Hank's balanced salt solution (HBSS) containing 0.5% fetal bovine serum (FBS), 10 mM EDTA, 1.5 mM dithiothreitol and 10 mM HEPES at 37 °C for 35 min with agitation to remove the epithelial cell layer. After the removal of the epithelial cells, tissues were washed in PBS, finely diced and incubated in HBSS buffer (w Ca/Mg)

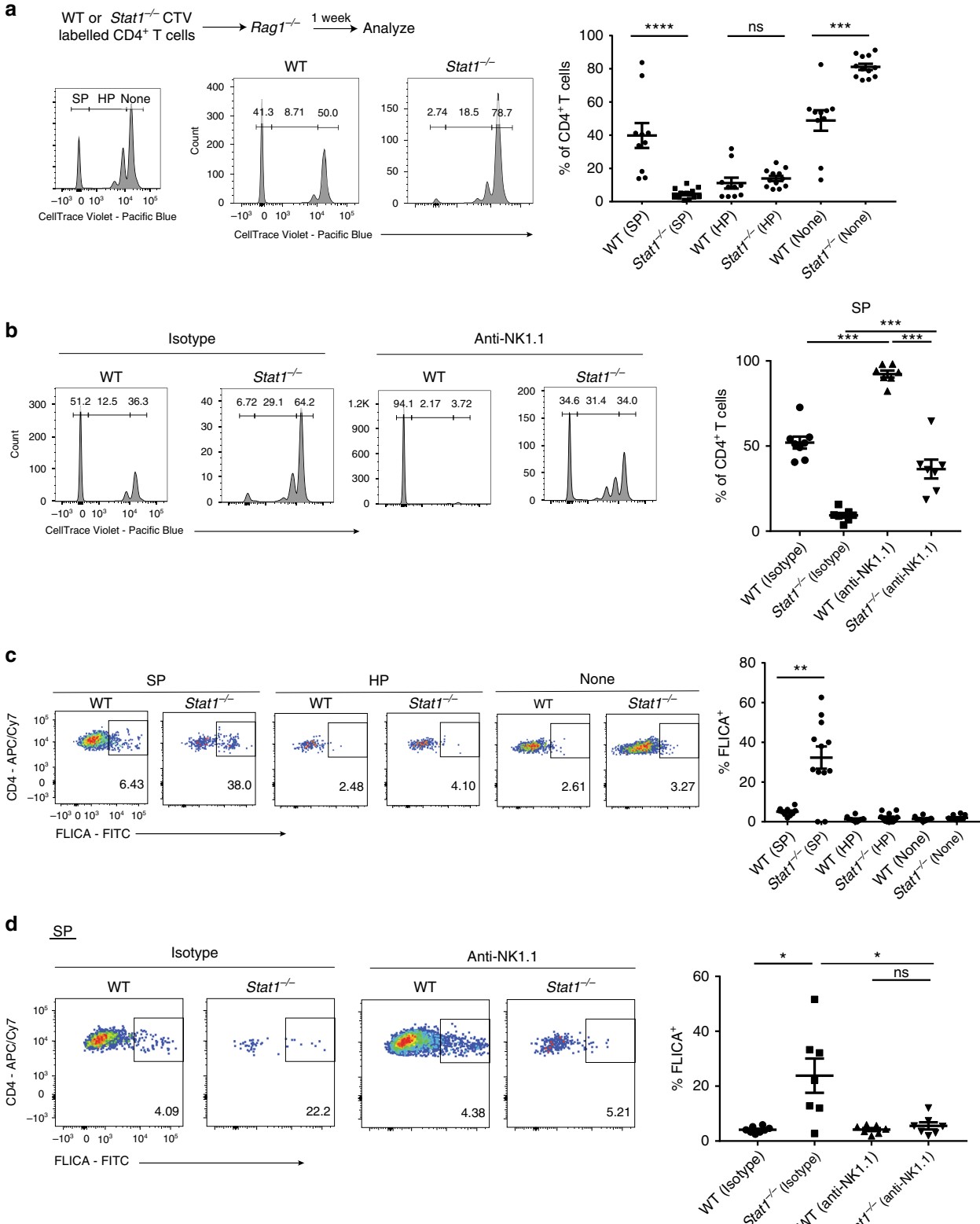

**Fig. 7** NK cells specifically eliminate $Stat1^{-/-}$ T cells undergoing spontaneous proliferation. $Rag1^{-/-}$ mice were injected with equal numbers ($3–4 \times 10^6$) of WT or $Stat1^{-/-}$ CTV labeled unfractionated CD4$^+$ T cells and analyzed after 1 week. **a** Schematic of experiment, as well as representative flow cytometry plots of T cells in the spleen + lymph nodes (gated on live CD45$^+$ CD3$\varepsilon^+$ CD4$^+$ cells, Supplementary Fig. 4d) followed by their mean frequencies ± SEM. **b**, **d** Similar to **a**, but with 400μg anti-NK1.1 antibody or Isotype Control injected 1 day prior to T-cell transfer. **b** CTV profiles of the T cells are shown as well as the mean frequencies ± SEM of the SP population. **c** Representative images of FLICA staining from the T cell SP, HP and non-proliferating populations in **a**, as well as their mean frequencies ± SEM. **d** FLICA staining in the SP population from **b** shown with mean frequencies ± SEM shown. Pooled from three to four independent experiments, with each point representing an individual mouse. *$p < 0.05$, **$p < 0.01$, ***$p < 0.001$, ***$p < 0.0001$ by two-tailed Mann–Whitney test

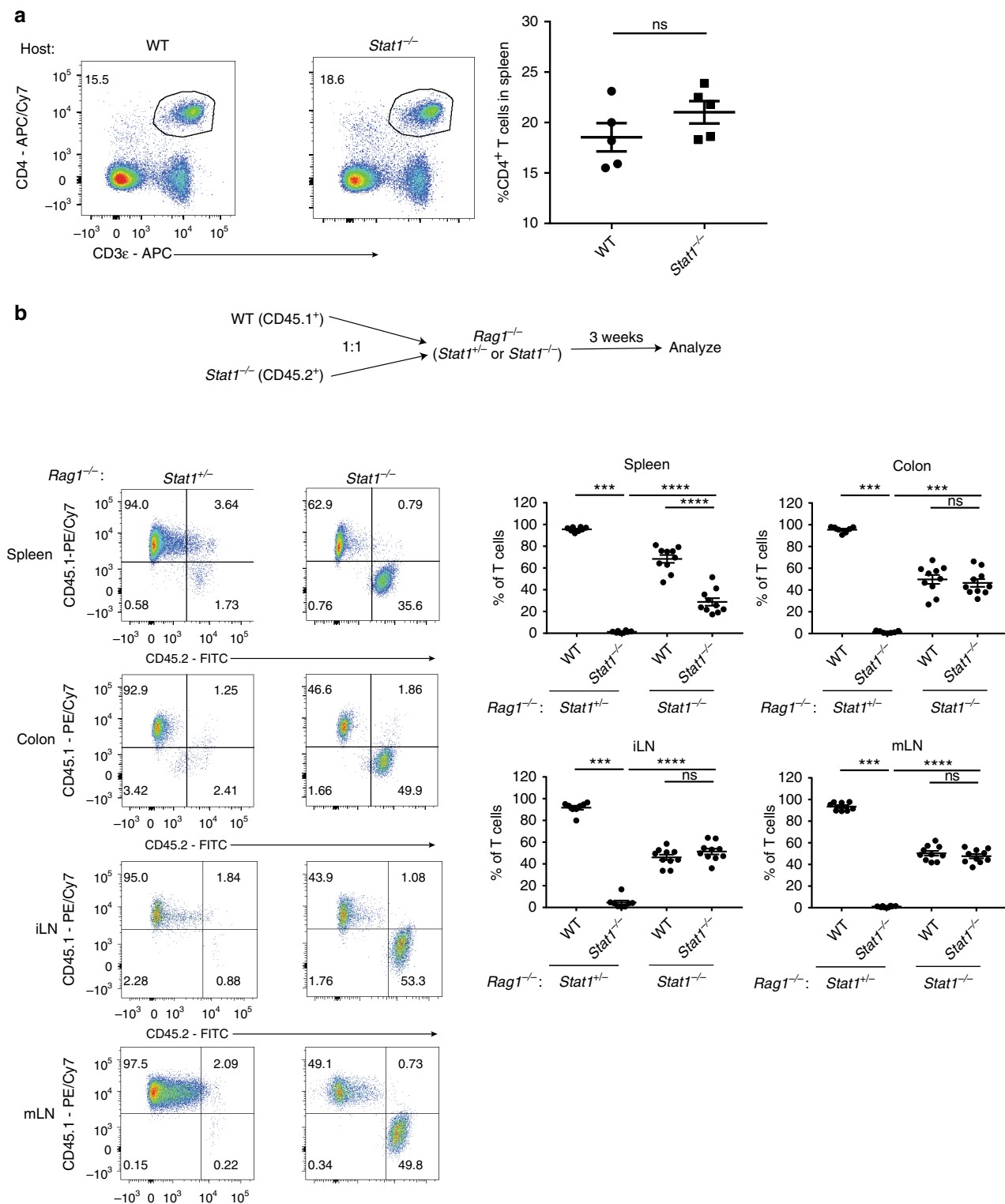

**Fig. 8** Innate STAT1 expression is required to eliminate *Stat1*$^{-/-}$ T cells. **a** Representative flow cytometry plots of CD4$^+$ T cells in the spleen of WT and *Stat1*$^{-/-}$ mice followed by their mean frequencies ± SEM. **b** WT (CD45.1$^+$) or *Stat1*$^{-/-}$ (CD45.2$^+$) CD4$^+$ T cells were injected i.p. at a 1:1 ratio (1 × 10$^6$/ type) into *Stat1*$^{-/-}$*Rag1*$^{-/-}$ mice or their *Stat1*$^{+/-}$*Rag1*$^{-/-}$ littermate controls and analyzed after 3 weeks. Representative images of CD45.1$^+$ vs CD45.2$^+$ cells (gated on live CD45$^+$ CD3$\varepsilon^+$ CD4$^+$ T cells) from various organs are shown followed by their mean frequencies ± SEM. Pooled from three independent experiments, with each point representing an individual mouse. ***$p < 0.001$, ****$p < 0.0001$ by two-tailed Mann–Whitney test

containing 20% FBS, 10 mM HEPES, 1.5 mM CaCl₂ and collagenase VIII (200 U/ml) at 37 °C for 40 min with agitation. Tissues were then repeatedly flushed through a 10 ml syringe and further incubated for 15 min. Digested tissues were filtered, washed in PBS and used for flow cytometry.

**In vitro T-cell proliferation.** Unfractionated CD4$^+$ T cells were first labeled with 5 μM CFSE for 5 min at room temperature and washed repeatedly with PBS containing FBS. They were then cultured in 96-well flat-bottom plates containing plate-bound anti-CD3ε (3 μg/ml, eBioscience) and soluble anti-CD28 (1 μg/ml, eBioscience) for 3 days. T cells were cultured in DMEM containing 10% FBS, L-glutamine, pyruvate, non-essential amino acids, MEM vitamins, L-arginine, L-asparagine, folic acid, β-mercaptoethanol and pen/strep.

**Reagents.** For flow cytometric staining, antibodies against the following were used (Clone name, dilution, manufacturer and catalog number in brackets): CD3ε (145-2C11, 1:300–400, Biolegend #100312/100306), TCRβ (H57-597, 1:400, Biolegend #109222), CD4 (GK1.5, 1:300 Biolegend #100414), NKp46 (29A1.4, 1:50, Biolegend #137604), CD49b (HMα2, 1:200 Biolegend #103517), CD45 (30-F11, 1:500, Biolegend #103140), H2-K$^b$/D$^b$ (28-8-6, 1:100, Biolegend #114606/114607), Qa-2 (695H1-9-9, 1:100, Biolegend #121709), Qa-1b (6A8.6F10.1A6, 1:10, Miltenyi Biotec #130-104-220), CD16/32 (93, 0.5 μg/10$^6$ cells, Biolegend #101302), IL-17A (TC11-18H10.1, 1:125, Biolegend 506904), IFNγ (XMG1.2, 1:200, Biolegend #505809/eBioscience #17-7311-82), Mouse IgG2a, κ Isotype Ctrl (MOPC-173, 1:100, Biolegend #400207/400211), Mouse IgG1, κ Isotype Ctrl (MOPC-21, 1:66.7, Biolegend #400119), CD45.1 (A20, 1:300, Biolegend #110729), CD45.2 (104, 1:300, Biolegend #109806). For T-cell stimulation, antibodies against CD3ε (145-2C11, 3 μg/ml, eBioscience #16-0031-86) and CD28 (37.51, 1 μg/ml, eBioscience #16-0281-85) were used. For NK depletion assays, antibodies against NK1.1 (PK136, BioXCell #BP0036) or Isotype Control (C1.18.4, BioXCell #BP0085) were used.

**Flow cytometry.** For flow cytometry and sorting experiments, cells were stained in flow cytometric staining buffer (2% FBS plus 0.1% NaN₃ in PBS) and MACS buffer (0.5% BSA and 2 mM EDTA in PBS), respectively. For antibody staining of surface markers, cells were incubated with anti-CD16/32 (Biolegend) for 10 min at room temperature to block Fc receptors, before being incubated with antibodies for 20–30 min at 4 °C. Cells were also incubated with Zombie Violet Fixable Viability Dye (1:400, Biolegend) or 7-AAD (1:20, BD Biosciences) according to the manufacturer's instructions to identify and exclude dead cells. For intracellular cytokine staining, cells were incubated with PMA (50 ng/ml), ionomycin (500 ng/ml), and GolgiStop (1:1000, BD Biosciences) for 4 h at 37 °C. After staining for surface markers, cells were fixed and permeabilized with Cytofix/Cytoperm (BD Biosciences), followed by staining in Perm/Wash buffer (BD Biosciences) according to the manufacturer's instructions. For assessment of cell death, cells were stained with the FAM-FLICA Poly Caspase Kit (ImmunoChemistry Technologies) for 1 h at 37 °C in T-cell media prior to antibody surface staining. All samples were acquired with a BD Canto II or LSRFortessa Flow Cytometer (BD Biosciences) and analyzed with FlowJo (FlowJo, LLC).

**RNA sequencing.** In the post-transfer setting, WT or Stat1$^{-/-}$ T cells (gated as CD45$^+$ CD3ε$^+$ CD4$^+$, Supplementary Fig. 2d) were FACS sorted from the spleen and lymph nodes of Rag1$^{-/-}$ mice post transfer directly into RLT lysis buffer (Qiagen) and RNA extracted using the RNeasy Micro kit (Qiagen). As Stat1$^{-/-}$ T cells showed reduced survival/expansion in vivo, it was not technically feasible to acquire sufficient cells for purity analysis by flow cytometry, hence purity was determined by confirming the downregulation of Stat1 in the Stat1$^{-/-}$ T cells. Library preparation, RNA-seq and analysis were performed at the Molecular Biology Core Facility (MBCF) of Dana-Farber Cancer Institute, Boston, using the Clontech SMARTer v4 kit for mRNA library generation and the Illumina NextSeq 500 Platform (Single-end 75 bp) for sequencing. The data was analyzed using the VIPER algorithm[59], with reads aligned to the mouse mm9 genome using STAR, transcripts assembled with Cufflinks and differential analysis performed with DESeq2. Gene Ontology analysis was performed using the PANTHER Over-representation test (http://www.geneontology.org/). The raw and processed data for RNA sequencing are deposited in the NCBI GEO database under GSE116475.

**Statistical analysis.** Statistical analyses were performed with GraphPad Prism software using two-way ANOVA with Bonferroni's multiple comparisons test, two-tailed Mann–Whitney test or two-tailed t-test as indicated in the figure legends. Significance was defined as p-value < 0.05 using the following notations: *$p < 0.05$, **$p < 0.01$, ***$p < 0.001$, ****$p < 0.0001$.

**Reporting summary.** Further information on experimental design is available in the Nature Research Reporting Summary linked to this article.

## Data availability
All data in this study are available from the corresponding author upon reasonable request. RNAseq data has been deposited in the GEO under GSE116475. A reporting summary for this Article is available as a Supplementary Information file, as well as a Source Data file with the source data underlying the weight curves in Figs. 1a, 3b, and 6b where individual data points are not displayed.

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

## Acknowledgements

We thank Jeremy Goettel for assistance in histological scoring, Ryan Kelly for technical assistance with the experiments and Jon Kagan, Ivan Zanoni and Bruce Horwitz for helpful discussions. We also thank the Dana-Farber Cancer Institute Flow Cytometry Core for cell sorting and the DFCI Molecular Biology Core Facilities - Genomics for RNA-seq and analysis. Y.H.K. is supported by an A*STAR National Science Scholarship, Singapore. A.B. is supported by the CCFA Career Development Award (327200) and a NIH KO1 award (K01DK109026). S.B.S. is supported by NIH Grants DK034854, and AI50950, the Helmsley Charitable Trust, and the Wolpow Family Chair in IBD Treatment and Research.

## Author contributions

Y.H.K. and S.B.S. conceived the study. Y.H.K., A.B., and S.B.S. designed the experiments. Y.H.K., M.F. and A.B. performed the experiments, acquired and analyzed the data. Y.H.K. and S.B.S. wrote the manuscript.

## Additional information

**Competing interests:** S.B.S. declares the following interests: Scientific advisory board participation for Pfizer, Janssen, Celgene, IFM therapeutics, and Pandion Inc. Grant support from Pfizer, Janssen, Merck. Consulting for Hoffman La Roche and Amgen. The remaining authors declare no competing interests.

