## [Peer Review File · Nature Communications]

Reviewers' comments:

Reviewer #1 (Cytokine signaling, IBD)(Remarks to the Author):

The manuscript entitled "STAT1 signaling shields T cells from NK cell-mediated cytotoxicity to enable intestinal inflammation" reports that STAT1 deficient T cells has a reduced level of Nr1h3 and MHC class I molecules, which are inhibitors of NK cell-mediated killing and thus fail to expand and establish colitis in lymphopenic mice. The experimental results are relatively clear and the research hypothesis is well proven.

1. What was the change in colitis phenotype (body weight, colon length, histologic analysis etc) when STAT1 deficient CD4 + T cells were transferred to Rag1 KO mice?
2. What is the phenotypic change of IL-10rb and Rag1 double KO mice according to age? The effect of IL-10 receptor-deficient NK cells on STAT1-deficient CD4 T cells was confirmed, so the role of IL-10 in NK cells should be discussed in the discussion section. And I wonder how the possibility of the interaction between STAT1 deficient CD4 + T cells and macrophages in IL-10rb and Rag1 double KO mice.
3. Does the STAT1 signal in the T cell affect the action of other cells besides the NK cell? The value recovered through anti-NK 1.1 treatment is not so large, and other roles through STAT1 in T cell may play a role.

Reviewer #2 (NK and T cell function, gut inflammation)(Remarks to the Author):

The authors have demonstrated that Stat1 and associated expression of MHC class I molecules would protect CD4+ T cells from elimination by natural killer cells, leading to intestinal inflammation. Although the manuscript is potentially interesting, it remains unclear where the deletion of Stat1 gene actually affects the survival of CD4+ T cells, since an effect of Stat1 may be directed toward both NK cell-mediated T cell deletion and homeostatic proliferation.

1. The data shown in the manuscript are obtained through intraperitoneal transfer of 1×10^6 cells of unfractionated CD4+ T cells into RAG-deficient mice, in which homeostatic proliferation of the transferred cells is thought to take place. Cytokines such as IL-7 and IL-15 are indispensable for such homeostatic expansion of naïve and memory T cells, and Stat1 is one of the signaling molecules downstream of IL-15 receptor. In addition, a recent report suggested that T cells undergoing lymphopenia-induced proliferation would upregulate Stat1 protein, which would trigger an alternate IL-7-dependent program. Therefore, it is possible that failed expansion of Stat1-deficient T cells may be due to lack of IL-7 or IL-15 signaling, rather than failure in NK cell-mediated elimination of the T cells. The authors should conduct more experiments to challenge this kind of criticism.
2. How long does it take to get a full reconstitution by transferred T cells? Is it possible to obtain similar results by using tamoxifen-induced deletion of Stat1 gene after full reconstitution of T cells? Again, it is not clear to me whether the authors really observe what they are arguing in the manuscript.
3. If a marked reduction of Stat1-deficient T cells was observed even in the absence of colonic inflammation, the title of the manuscript is misleading. Colonic inflammation is just a result of manipulating animals under the specific experimental setting.
4. What are the physiological or pathological relevance of the findings? An experiment such as transfer of irrelevant T cells into RAG-deficient mice should be considered.

Point by Point Critique:

Essentials from the editor:

"We ask that you include additional empirical data to test the effects of STAT1 deficiency on CD4 T homeostatic expansion in the context of adoptive cell transfer colitis, as requested by referee #2. We also ask the other raised points by both referees to be addressed."

We agree with the editor and reviewer 2's comments and have tested the role of STAT1 deficiency in homeostatic expansion. The data is provided in the manuscript and is explained in reviewer 2's comment below.

Reviewers' comments:

Reviewer #1:

The manuscript entitled "STAT1 signaling shields T cells from NK cell-mediated cytotoxicity to enable intestinal inflammation" reports that STAT1 deficient T cells has a reduced level of *Nlrc5* and MHC class I molecules, which are inhibitors of NK cell-mediated killing and thus fail to expand and establish colitis in lymphopenic mice. The experimental results are relatively clear and the research hypothesis is well proven.

1. What was the change in colitis phenotype (body weight, colon length, histologic analysis etc) when STAT1 deficient CD4 + T cells were transferred to *Rag1* KO mice?

Thank you for the important comment. These mice do not have a colitic phenotype as evidenced by the lack of weight loss, colonic thickening and histological inflammation. See below.

Figure R1

2a. What is the phenotypic change of IL-10^{rb} and *Rag1*

double KO mice according to age?

We do not see colitis in *Il10rb*^{-/-}*Rag1*^{-/-} mice if no T cells are transferred (Shouval et al. *Immunity* 2014). The T cell transfer experiments are typically done in > 8 week old mice, and we do not observe a correlation between the age of these mice and the degree of colitis induced by WT T cells.

Il10rb^{-/-} mice however do spontaneously develop colitis in an age-dependent manner (i.e. from weaning age), and we refer the reviewer to our earlier published work (Redhu et al. *eLife* 2017).

2b. The effect of IL-10 receptor-deficient NK cells on STAT1-deficient CD4 T cells was confirmed, so the role of IL-10 in NK cells should be discussed in the discussion section.

We have updated the discussion section accordingly.

2c. And I wonder how the possibility of the interaction between STAT1 deficient CD4+ T cells and macrophages in IL-10rb and Rag1 double KO mice.

We agree with the reviewer that macrophages can interact with *Stat1*^{-/-} T cells, as it has been shown previously that they can phagocytose $\beta 2m^{-/-}$ tumors (Barkal et. al. Nat Immunol 2017), and have preliminary data addressing this concern in the 3rd comment below.

3. Does the STAT1 signal in the T cell affect the action of other cells besides the NK cell?

The value recovered through anti-NK 1.1 treatment is not so large, and other roles through STAT1 in T cell may play a role.

We agree with the reviewer that other cell types might play a role in the elimination of *Stat1*^{-/-} T cells as we do not observe a complete rescue of these cells compared to WT in NK depleted mice. We have done preliminary experiments depleting several cell types in addition to NK cells and transferring CellTrace Violet (CTV) labelled *Stat1*^{-/-} T cells to see if we can get further enhancement of its expansion 2 weeks after T cell transfer.

We explored the role of macrophages (explained in the 2nd comment above) by using an anti-CSF1R blocking antibody, which effectively reduced splenic red pulp macrophages (Fig. R2a). Compared to NK-depleted *Rag1*^{-/-} mice that received the control antibody, the depletion of macrophages did not lead to an increase in T cell levels (Fig. R2b).

Fig. R2. *Stat1*^{-/-} CellTrace Violet (CTV) labelled T cells were injected i.p. into NK-depleted *Rag1*^{-/-} mice that were treated with control or anti-CSF1R antibody, or *Rag2*^{-/-}*I2rg*^{-/-} mice treated with control antibody. **(a)** Representative flow cytometry plots of splenic red pulp macrophages (CD11b^{lo}F4-80⁺, Hashimoto et. al. J Exp Med 2011) shown at 2 weeks post T cell transfer, with their mean frequencies \pm SEM. **(b)** CD4⁺ T cell percentages and total numbers in the spleen (mean frequencies \pm SEM) are shown 2 weeks post T cell transfer. **(c)** Representative CTV plots of T cells shown, along with the level of spontaneous proliferation (CFSE^{lo}) at 2 weeks post T cell transfer (mean frequencies \pm SEM). *Rag2*^{-/-}*I2rg*^{-/-} mice were cohoused for at least 2 weeks with *Rag1*^{-/-} mice to control for the microbiota. Macrophage depletion was achieved by i.p. injection of 400 μ g Isotype or anti-CSF1R antibody (BioXCell) twice weekly beginning 1 week before T cell transfer. NK depletion was achieved by i.p. injection of 400 μ g anti-NK1.1 antibody into *Rag1*^{-/-} mice 1 day before T cell transfer, as well as 200 μ g 1 week after to maintain depletion.

We also explored the role of ILCs using *Rag2^{-/-}Il2rg^{-/-}* mice, which naturally also do not have NK cells. We also do not see an enhancement of *Stat1^{-/-}* T cell expansion in this setting (Fig. R2b). However, the CTV profile suggested that the T cells display less proliferation compared to the NK-deplete *Rag1^{-/-}* mice. While this suggests that ILCs might promote the expansion of *Stat1^{-/-}* T cells, further work is needed to address other factors such as γ_c signaling on innate immune cells and stromal cells, as well as slight differences in the genetic background (*Rag2^{-/-}Il2rg^{-/-}* mice are B6;10 vs *Rag1^{-/-}* which are B6).

All in all, these data suggest that CSF1R-dependent macrophages and ILCs are not the reason behind the incomplete rescue of *Stat1^{-/-}* T cells. It is possible that other cell types that we've not looked at mediate this difference. It is also possible that in addition to protecting T cells from NK cell killing, STAT1 signaling itself might promote T cell proliferation *in vivo* downstream of cytokines other than Type I+II IFN, such as IL-6, which is discussed in the manuscript.

Reviewer #2.

The authors have demonstrated that Stat1 and associated expression of MHC class 1 molecules would protect CD4+ T cells from elimination by natural killer cells, leading to intestinal inflammation.

Although the manuscript is potentially interesting, it remains unclear whether the deletion of Stat1 gene actually affects the survival of CD4+ T cells, since an effect of Stat1 may be directed toward both NK cell-mediated T cell deletion and homeostatic proliferation.

1. The data shown in the manuscript are obtained through intraperitoneal transfer of 1x10⁶ cells of unfractionated CD4+ T cells into RAG-deficient mice, in which homeostatic proliferation of the transferred cells is thought to take place. Cytokines such as IL-7 and IL-15 are indispensable for such homeostatic expansion of naïve and memory T cells, and Stat1 is one of the signaling molecules downstream of IL-15 receptor. In addition, a recent report suggested that T cells undergoing lymphopenia-induced proliferation would upregulate Stat1 protein, which would trigger an alternate IL-7-dependent program. Therefore, it is possible that failed expansion of Stat1-deficient T cells may be due to lack of IL-7 or IL-15 signaling, rather than failure in NK cell-mediated elimination of the T cells. The authors should conduct more experiments to challenge this kind of criticism.

We thank the reviewer for this important comment, and provide data addressing this issue that is in the updated manuscript (Figure 7). The data is briefly explained below.

Here, we transferred CellTrace Violet labelled WT or *Stat1^{-/-}* T cells into *Rag1^{-/-}* mice and looked at their proliferation earlier at 1 week post transfer to visualize sufficient numbers of *Stat1^{-/-}* T cells. Earlier reports have shown that T cells in this scenario undergo slow Homeostatic expansion (HP) that is driven by IL-7/15 as well as fast spontaneous expansion (SP) that is driven by the microbiota and IL-6. We show that *Stat1^{-/-}* T cells display a reduction specifically in the SP population and not the HP population. This argues against a role for IL-7/15 in providing the STAT1 dependent signal as these regulate HP. Moreover, we show that the defect in *Stat1^{-/-}* SP is rescued by NK cell depletion, thus showing a role for NK cells in restricting *Stat1^{-/-}* T cells that undergo SP.

We also addressed the survival of the T cells by assaying for cell death in the SP/HP/non-proliferating populations using FAM-FLICA that stains for activated caspases. This was challenging as *Stat1^{-/-}* T cells have a dramatically reduced SP population. Nevertheless, we observed increased cell death in *Stat1^{-/-}* T cells specifically in the SP population and not the HP or non-proliferating population, and this increased SP cell death is reversed upon NK cell depletion. Overall, these data strongly point to a role for NK cells in killing *Stat1^{-/-}* SP T cells, which are then subsequently cleared.

We note that the degree of rescue of SP in *Stat1^{-/-}* T cells is incomplete, which correlates with the incomplete rescue of T cell levels and the colitis in *Il10rb^{-/-}Rag1^{-/-}* mice. We are unclear as to the reason behind this. It is possible that there might be other cell types involved in clearing *Stat1^{-/-}* T cells, although NK cell depletion is sufficient to reverse the cell death phenotype in *Stat1^{-/-}* T cells. Alternatively, IL-6, which drives SP, also activates STAT1, but this might not be the sole factor as a

recent report showed that IL-6 regulates T cell expansion only during inflammation. We refer the reviewer to the manuscript for a more detailed discussion.

2. How long does it take to get a full reconstitution by transferred T cells? Is it possible to obtain similar results by using tamoxifen-induced deletion of Stat1 gene after full reconstitution of T cells? Again, it is not clear to me whether the authors really observe what they are arguing in the manuscript.

We are unfortunately unable to obtain commercially available *Cre-ERT2 Stat1^{fl/fl}* mice or labs which have published with these mice which will directly address the reviewer's question within a reasonable time frame. We do however understand the reviewer's underlying concern of whether T cell specific STAT1 deletion in lymphoreplete mice also leads to its elimination.

To address the concern, we transferred WT or *Stat1^{-/-}* T cells (CD45.2⁺) into WT CD45.1 mice to see if *Stat1^{-/-}* T cells are also reduced when there's no lymphopenia. Consistent with earlier reports, there is reduced overall proliferation when T cells are transferred into lymphoreplete mice compared to lymphopenic mice. We do not observe a reduction in overall T cell percentages when *Stat1^{-/-}* T cells are transferred into lymphoreplete mice (Figure R3a). However, in the SP population, there is a mild but consistent decrease when T cells lack STAT1 (Figure R3b). This supports the earlier data showing that NK cells specifically target *Stat1^{-/-}* T cells undergoing SP. We thus believe that should STAT1 be inducibly deleted after reconstitution, they would not be eliminated unless there's a trigger for SP.

Fig. R3. WT CD45.1 mice were injected i.p. with 2×10^6 unfractionated WT or *Stat1^{-/-}* CTV labelled CD4⁺ T cells. **(a)** Representative flow cytometry plots of CD45.2⁺ cells in the spleen + lymph nodes (gated on live CD45⁺ CD3 ϵ ⁺ CD4⁺ T cells) and their mean frequencies \pm SEM at 1 week post transfer. **(b)** Representative flow cytometry plots of CTV profiles in CD45.2⁺ cells in the spleen + lymph nodes and the mean frequencies \pm SEM of SP, HP and non-proliferating populations at 1 week post transfer. Data pooled from 2 independent experiments. * $p < 0.05$ by Mann-Whitnev.

3. If a marked reduction of *Stat1*-deficient T cells was observed even in the absence of colonic inflammation, the title of the manuscript is misleading. Colonic inflammation is just a result of manipulating animals under the specific experimental setting.

We agree with the reviewer that the primary effect of STAT1 is to promote T cell survival with colonic inflammation being a secondary effect and will alter the title to reflect this emphasis.

We do however believe that this mechanism is relevant for intestinal immune disorders, as we now show that NK cells specifically target *Stat1^{-/-}* T cells in the setting of SP which has been reported by

other groups to be driven by the microbiota. We think that the reason why *Rag1*^{-/-} mice don't develop colitis with unfractionated CD4⁺ T cells is likely due to the presence of Tregs.

4. What are the physiological or pathological relevance of the findings? An experiment such as transfer of irrelevant of T cells into RAG-deficient mice should be considered.

The T cell transfer model of colitis that we've utilized is a relevant and robust model that has been utilized by several groups to understand IBD pathogenesis. As an example, we've used this model to uncover a role for IL-1 in T cell driven colitis in *Il10rb*^{-/-}*Rag1*^{-/-} mice, which led us to utilize anakinra as a therapeutic approach in patients with Il-10R deficiency. (*Shouval et. al. Gastroenterology 2016*)

With regards to the findings of STAT1 deficiency, we believe this is important as several groups have utilized the IBD model, as well as other similar models (GvHD, Autoimmunity) to draw conclusions about the role of STAT1 in T cell driven disease without examining the role of NK cells, as described in the discussion section of the manuscript. Moreover, our finding that STAT1 specifically protects T cells that undergo the microbiota driven SP provides a mechanism by which NK cells regulate commensal driven T cell responses. We note that earlier reports showed that *Ifnar1*^{-/-} antiviral T cells are eliminated by NK cells only in the setting of viral infection, and our work adds to this paradigm by showing that this mechanism is relevant in regulating microbiota driven T cell responses, which will be helpful in studying intestinal disorders such as IBD. It also suggests that, therapeutically, one can consider specifically target STAT1 in T cells as a way of reducing commensal driven T cell inflammation.

Finally, STAT1's intrinsic antiproliferative activity has led scientists to consider targeting STAT1 for improving T cell reconstitution in clinical lymphopenia e.g. HIV infection, transplantation. Our findings suggest that this approach will have to be balanced with ensuring sufficient MHC-I levels to prevent NK cell mediated T cell depletion.

With regards to the reviewer's comments about transferring irrelevant T cells into RAG deficient mice, several groups have published work studying the role of TCR in T cell SP and HP by using T cells with genetically defined TCRs such as OT-I/II (e.g. *Kieper et. al. J Immunol 2005*, *Feng et. al. J Exp Med 2010*). The use of these irrelevant T cells was reported to lead to reduced overall SP, which is analogous to our data in lymphoreplete mice where there is also reduced overall SP. Thus, we believe that performing this experiment would not yield significant insight over the previous data (Figure R3).

REVIEWERS' COMMENTS:

Reviewer #1 (Remarks to the Author):

Authors responded to my comments faithfully. Most of the concerns were adequately answered.

Reviewer #2 (Remarks to the Author):

The authors have responded to my requests and revisions appear to be appropriate.